# Neuroendocrine Neoplasms of the Gynecologic Tract

**DOI:** 10.3390/cancers14071835

**Published:** 2022-04-06

**Authors:** Mayur Virarkar, Sai Swarupa Vulasala, Ajaykumar C. Morani, Rebecca Waters, Dheeraj R. Gopireddy, Sindhu Kumar, Priya Bhosale, Chandana Lall

**Affiliations:** 1Department of Diagnostic Radiology, University of Florida College of Medicine, 655 West 8th Street, C90, 2nd Floor, Clinical Center, Jacksonville, FL 32209, USA; vulasalaswarupa@gmail.com (S.S.V.); dheerajreddy.gopireddy@jax.ufl.edu (D.R.G.); sindhu.kumar@jax.ufl.edu (S.K.); chandana.lall@jax.ufl.edu (C.L.); 2Department of Diagnostic Radiology, The University of Texas MD Anderson Cancer Center, 1515 Holcombe Blvd., Houston, TX 77030, USA; amorani@mdanderson.org (A.C.M.); priya.bhosale@mdanderson.org (P.B.); 3Department of Pathology, The University of Texas MD Anderson Cancer Center, 1515 Holcombe Blvd., Houston, TX 77030, USA; rwaters@mdanderson.org

**Keywords:** gynecological NENs, neuroendocrine, cervical neuroendocrine tumor, FIGO classification, imaging of neuroendocrine tumors, PET/CT

## Abstract

**Simple Summary:**

Neuroendocrine refers to the cells that synthesize and secrete messenger chemicals such as neuropeptides and amines. Neuroendocrine neoplasms (NENs) are aggressive tumors arising from neuroendocrine cells, with an annual incidence of 6.98/100,000 and a prevalence of 170,000 in the United States. Primary gynecologic NENs constitute ≤2% of female reproductive tumors. NENs of the gynecologic tract are associated with high recurrence rates and dismal prognosis, making their treatment challenging. This article focuses on the updated staging classifications, clinicopathological characteristics, imaging, and management of NENs of the gynecological tract.

**Abstract:**

Gynecological tract neuroendocrine neoplasms (NEN) are rare, aggressive tumors from endocrine cells derived from the neuroectoderm, neural crest, and endoderm. The primary gynecologic NENs constitute 2% of gynecologic malignancies, and the cervix is the most common site of NEN in the gynecologic tract. The updated WHO classification of gynecologic NEN is based on the Ki-67 index, mitotic index, and tumor characteristics such as necrosis, and brings more uniformity in the terminology of NENs like other disease sites. Imaging plays a crucial role in the staging, triaging, restaging, and surveillance of NENs. The expression of the somatostatin receptors on the surface of neuroendocrine cells forms the basis of increasing evaluation with functional imaging modalities using traditional and new tracers, including ^68^Ga-DOTA-Somatostatin Analog-PET/CT. Management of NENs involves a multidisciplinary approach. New targeted therapies could improve the paradigm of care for these rare malignancies. This article focuses on the updated staging classifications, clinicopathological characteristics, imaging, and management of gynecologic NENs of the cervix, ovary, endometrium, vagina, and vulva, emphasizing the relatively common cervical neuroendocrine carcinomas among these entities.

## 1. Introduction

Neuroendocrine neoplasms (NENs) are rare tumors with an annual incidence of 6.98/100,000 and a prevalence of 170,000 in the United States [1,2,3]. They were discovered in the late 19th century by Langhans and Lubarsch [4]. They include a heterogeneous group of neoplasms from the endocrine cells derived from the neuroectoderm, neural crest, and endoderm [4,5]. Neuroendocrine refers to cells that synthesize and secrete messenger chemicals, such as neuropeptides and amines [4]. Neuroendocrine cells express somatostatin receptors on the surface, forming functional imaging modalities, including ^68^Ga-DOTA-Somatostatin Analog-PET/CT. In addition, they express neuroendocrine markers such as synaptophysin and chromogranin A, which are stored in vesicles and granules, respectively [6]. The most frequent locations of NENs are the gastrointestinal tract (62–70%), followed by the tracheobronchial system (25%) [4,6,7].

Primary gynecologic NENs constitute ≤2% of female reproductive tumors [8]. In a study by Crane et al., the origin of these neoplasms was 54% in the cervix, 24% in the uterine corpus, 16% in the ovary/fallopian tube, 5% in the vagina, and 1% in the vulva of the study participants [9]. Along the gynecologic tract, the ovary is the most common site for low-grade NETs. In contrast, the cervix is the most common site for aggressive, poorly differentiated neuroendocrine carcinoma (NEC) [10,11,12]. The presenting symptoms are non-specific and depend on the organ of origin. Gibbs et al. reported the extrauterine spread in 83.6% of uterine, 83.5% of ovarian, and 67% of cervical NENs upon diagnosis [13]. A few patients may manifest paraneoplastic syndromes due to the secretion of peptides from these neoplasms [14]. Overall, NENs of the gynecologic tract are uncommon and aggressive, with high recurrence rates and dismal prognosis, making their treatment challenging. Imaging evaluation of these neoplasms is similar to other histologic types, with added benefits of functional somatostatin receptor (SSTR) imaging. There are no reliable guidelines for either the diagnosis or management of these cases. The revised International Federation of Gynecology and Obstetrics (FIGO) Staging System remains the preferred staging system for gynecologic NENs [14].

The article focuses on the updated staging classifications, clinicopathological characteristics, imaging, and management of NENs of the cervix, ovary, endometrium, vagina, and vulva, emphasizing the relatively common cervical NECs among these entities.

## 2. Nomenclature

In the early 20th century, NENs were termed carcinoids by Oberndorfer, which refers to their less-aggressive behavior with small and multiple foci of undifferentiated cellular formations, well-defined borders, and absent metastatic potential [4,15]. Later, the nomenclature was subjected to numerous revisions until the updated WHO terminology. A dedicated consensus conference was held in November 2017 at the International Agency for Research on Cancer (IARC), Lyon, under the auspices of the World Health Organization (WHO) Classification of Tumors Group. The consensus meeting framed the uniform classification for NENs, published in the 2018 WHO classification guide. The updated fifth edition WHO classification of gynecologic NEN is delineated in Table 1 and in-general grading criteria for NENs in Table 2 [8]. Grading of tumors is based on the Ki-67 index, mitotic index, and tumor characteristics such as necrosis [16]. In terms of gynecologic tract NENs, grades 1 and 2 represent well-differentiated low-grade NETs, while grade 3 represents poorly differentiated high-grade NECs. Grading is only performed for NETs (grades 1 and 2). Although there are some reports of the entity, NET G3 is still not officially included in the recently updated WHO classification [17], in contrast to the new NET G3 category for gastroenteropancreatic (GEP) NENs [18]. There is also no further differentiation of grade 1 versus grade 2 NETs in the ovaries. NENs of the ovary are classified into ovarian carcinoids (grade 1) and ovarian NEC (grade 3). Except for the ovaries, the terms carcinoid and atypical carcinoid were removed from the nomenclature; instead, these correspond to NETs in the updated classification [8,19].

## 3. Histology and Immunohistochemistry of NEN

Well-differentiated NETs comprise small uniform cells in nesting, trabecular, or gyriform/serpentine growth patterns. They demonstrate round to oval nuclei, inconspicuous nucleoli, fine to coarsely granular chromatin described as “salt and pepper”, and a pale to moderately eosinophilic cytoplasm [7,10]. NEC tumor cells are pleomorphic, which proliferate in a “sheet-like” pattern and contain irregular nuclei, high mitotic index, a few cytoplasmic secretory granules, and necrosis. Although NENs have characteristic histology, immunohistochemistry is essential to confirm the diagnosis. Neuroendocrine biomarkers, including synaptophysin, chromogranin A, CD56, CD57, neuron-specific enolase, synaptic vesicle protein2 (SV2), and protein gene product 9.5 (PGP 9.5), are used to identify NENs. The positive staining for these markers is observed in one or more NEN variants. The majority of the studies on the effectiveness of biomarkers are performed on GEP-NEN. An increased chromogranin A level is a widely accepted biomarker with 60–90% sensitivity. However, it is non-specific (specificity 10–35%), expressed in healthy tissues, inflammatory diseases, and other tumors such as prostate cancer, small cell lung cancer, and pancreatic cancer [20]. A high cut-off level of chromogranin A is considered to overcome these limitations. A study by Campana et al. reported that the chromogranin A level of 84–87 U/l has a sensitivity and specificity of 55% and 95%, respectively [21]. Elevated neuron-specific enolase ranging from 154–370 U/l has been observed in the literature depending on the stage of the disease and the tested subjects [22,23]. Studies have reported that the levels of neuron-specific enolase were decreased with treatment, yet recurrences were observed within a year [19,24,25]. The sensitivity and specificity of neuron-specific enolase in differentiating neuroendocrine from non-neuroendocrine tumors are 39–43% and 65–73%, respectively [26]. Although elevated neuron-specific enolase may indicate poorly differentiated tumors, it is no longer routinely used in clinical practice, as it is not superior to chromogranin A [27,28].

Nevertheless, no biomarker is highly accurate in determining the diagnosis and prognosis of the NEN. Hence a consensus on biomarkers for NEN suggested that a combination of imaging and circulating biomarkers should be considered to provide additional information on tumor behavior. Table 3 describes an overview of the imaging and circulating biomarkers studied in GEP-NEN [26,29]. However, the linear correlations between the imaging, biomarker, and final tissue-derived data have not been established [29]. The miRNAs are the non-coding RNAs that regulate the gene expression by degrading mRNAs or by inhibiting the translation [30]. The consensus by Oberg et al. also concluded that the multianalyte biomarkers such as mRNA have the highest sensitivity and specificity in identifying the minimal disease, predicting the efficacy of the treatment and prognosis (Figure 1) [29].

Immunohistochemistry can help to distinguish between primary and metastatic NENs. For instance, positive thyroid transcription factor-1 staining indicates the primary NEN in the thyroid and lung, ISL-1, and PDX-1 in the pancreas, PAX8 in the thyroid, and CDX-2 and villin in the gastrointestinal tract [11,31]. Recently, a new marker, insulinoma-associated protein 1 (INSM1), was reported in 90% of patients with high-grade genitourinary NEC on immunostaining by Chen et al. [32]. The sensitivity of INSM1 (90%) was similar to chromogranin (87%), synaptophysin (92%), and CD56 (85%), while the specificity was 97.4% in the report by Chen et al. [32]. According to Zou et al., the specificity of INSM1 in high-grade gynecologic NEC is 95% compared to chromogranin (82%), synaptophysin (81%), and CD56 (75%) [33]. Figure 2 illustrates the different grades and classifications of NENs.

## 4. Cervical NEN

The cervix is the most common site of NENs in the gynecologic tract, and 0.9–1.5% of cervical neoplasms belong to the NEN category [34,35] (Figure 3 and Figure 4). Around 200 cases of cervical NECs are diagnosed every year [36]. Cervical NENs is classified into cervical NETs and high-grade cervical NECs based on Ki-67 and the mitotic index. Low-grade (G1) tumors have ≤3 Ki-67 and 2/10 HPF mitotic indices, and include typical carcinoids, while intermediate-grade (G2) NETs (Ki-67 >3% and mitotic index 2–20/10HPF) include prior atypical carcinoid variants. In contrast, high-grade NECs have >20 Ki and >20/10 HPF mitotic indices, and include small and large cell NECs [8]. The small cell variant is the most common (80%) variant among cervical NENs, followed by large-cell NEC (12%) and differentiated (G1–G2) cervical NETs. Only a few cases of lower-grade cervical NETs have been reported in the literature [35]. NECs may also co-exist with adenocarcinoma or squamous cell carcinoma, and the clinical behavior of such cases is determined by the neuroendocrine component [37].

The incidence of cervical NENs peaks around the fourth decade of life. Patients may present with abnormal vaginal bleeding (22–73%), vaginal discharge, post-coital bleeding, and pelvic pain/pressure (8%) [35]. Given the aggressive nature of the neoplasm and extensive lymphatic invasion (37–57%), distant metastasis can be expected at the initial presentation. For example, stage IB1 cervical NEC is associated with positive pelvic lymph nodes in 40% of cases compared to 10–15% with the same stage (IB1) cervical squamous cell carcinomas [36]. The reported correlation between human papillomavirus (HPV) infection and cervical NECs in the literature prompts considering a prophylactic HPV vaccine to prevent NECs [36,38]. In a study by Castle et al., HPV was associated with 85% of small-cell and 88% of large-cell cervical NECs; of which, HPV-16 was detected in 10% and 29%, and HPV-18 was detected in 51% and 30% of small cell and large cell cervical NECs, respectively [39]. 

### 4.1. Molecular Characterization and Immunohistochemistry of Cervical NEC

At a molecular level, cervical NECs is associated with mutations in the PIK3CA gene (18%), KRAS gene (14%), and TP53 (11%) [40]. Hillman et al. reported that cervical NECs are genetically more similar to non-NEC cervical cancers than extra-cervical NECs of the lungs and bladder [41]. The coexistence of the PIK3CA mutation can explain this similarity in one of the APOBEC motifs in cervical NEC and non-NEC cancers [41,42]. Identifying genetic alterations in small-cell cervical NEC allows for the application of targeted therapies, including angiogenesis inhibitors, cell signaling pathway inhibitors, apoptosis promoters, and immunotherapies in managing small-cell cervical NEC. Xing et al. reported that the targeted next-generation sequencing of small-cell cervical NECs showed genetic alterations in the MAPK, TP53/BRCA, and PI3K/AKT/mTOR pathways. Such alterations allowed the tumor to be amenable to targeted therapies [43]. Similarly, a study by Cho et al. described recurring mutations in ATRX, EBRR4, and AKT/mTOR pathway genes in the pathogenesis of small-cell cervical NEC [44]. Schultheis et al. characterized genomes and discovered highly recurring Tp53 and Rb1 alterations in lung NEC and p53 and RB tumor suppressor protein inactivation in cervical NEC, indicating small-cell NECs are convergent phenotypes [45]. 

Immunohistochemistry has been implemented as an ancillary tool to diagnose high-grade cervical NEC. In a study by McCluggage et al., PGP9.5, chromogranin, synaptophysin, and CD56 were positive in 43%, 57%, 90%, and 90%, respectively, of cervical NEC [46]. CD56 and synaptophysin are the most sensitive markers; however, CD56 lacks specificity due to its wide occurrence in cervical neuroendocrine and non-neuroendocrine neoplasms [46]. Chromogranin is the most specific marker for cervical NEN [8]. Other markers studied in the literature include TTF1, p16, and insulinoma-associated protein-1 (INSM1). McCluggage et al. reported 71% positivity of TTF1 in cases of cervical NEC [46]. According to Maria et al., 86% of patients with cervical NEN overexpressed p16 upon immunostaining, in concordance with Alejo et al. [38]. Overexpression of p16, a cyclin-dependent kinase inhibitor, serves as a biomarker of HPV infection, with 97% of cervical squamous cell carcinomas and 50% of adenocarcinomas staining positive for p16 [36,38]. Hence, p16 cannot be relied on to make an NEN diagnosis. To distinguish from cervical squamous cell carcinoma, p63 staining is more valuable, as most cervical NENs stain negative for p63. Evolving markers such as insulinoma-associated protein-1 (INSM1) are noted to be specific, with positive staining in >95% of cases [47].

### 4.2. Prognostic Factors

NENs are associated with a worse prognosis than non-NEN cervical malignancies [48]. The most frequently related prognostic factors for early-stage disease are FIGO stage [49,50,51,52], lymph node metastasis [49,52,53,54], parametrial invasion, tumor size [55,56,57,58], lymphovascular space involvement, histological heterogeneity, older age [56,59,60,61,62], and smoking [56,63,64]. Of all these, Ishikawa et al. reported LVSI as the most significant prognostic factor for overall and disease-free survival. At the same time, pelvic lymph node metastasis was the critical prognostic factor for disease-free survival [63]. Early-stage disease without lymph node metastasis is a favorable prognostic factor [65]. Disease recurrence was noted in 50% of early-stage small cell cervical NEC patients included in a study by Pei et al. Distant organs were involved in most of the recurrences [66]. Hence, the study recommended adjuvant systemic chemotherapy for early-stage small cell cervical NEC.

Several studies show that advanced age is a poor prognostic factor, regardless of the tumor stage. Chen et al. reported that ages of ≤45 and >45 years are associated with overall survival of 62.5% and 35%, respectively (*p* = 0.04) [56]. Age of >60 years was associated with decreased survival in early-stage disease in a study by Intaraphet et al. (HR = 4.9; *p* = 0.007) (45), whereas the overall survival for advanced-stage disease were determined by both age at diagnosis (age > 60 years: HR = 9.9; *p* < 0.001; versus age < 45 years: HR = 3.4; *p* = 0.035) and FIGO stage IV (HR = 7.4; *p* = 0.024) [61]. Lin et al. reported that the threshold age of 50 years determines the overall survival in small cell cervical NEC (HR: 1.561; *p* = 0.034) [67]. Similarly, age < 50 years was associated with prolonged overall survival in a study by Hoskins et al. (*p* = 0.02) [68].

FIGO staging is an essential independent prognostic criterion for cervical NEC. The five-year overall survival rate is 20–50% for stage I–II and 2–15% for stage III–IV tumors [8]. In a study by Chen et al., the five-year overall survival rates of stages I–IV cervical NEC were 75%, 56%, 41%, and 0%, respectively, and the 5-year progression-free survival rates were 64%, 54.5%, 31%, and 0%, respectively [56]. Specific to small-cell cervical NEC, Wang et al. reported a five-year survival rate of 52% and 25% in stages I–IIA and IIB–IVB, respectively [66,69]. In a similar study by Intaraphet et al., the five-year cancer-specific survival was better in early-stage compared to advanced-stage disease (62.6% vs. 18.1%; *p* < 0.001) [61] in patients with cervical small cell NECs.

Tumors < 2 cm size have been found to be associated with better survival rates than the tumors > 2 cm (HR: 1.92; *p* = 0.055) [64]. In a study by Sheets et al., tumors < 2 cm had a more prolonged progression-free survival rate than tumors > 2 cm [70]. Chen et al. described that tumors < 4 cm size exhibit better overall survival than tumors ≥ 4 cm (76% vs. 39%; *p* = 0.013) [56]. Similarly, larger tumor size (≥4 cm) was associated with poor prognosis in a study by Liao et al. (HR: 2.4; *p* = 0.006) [58], and tumors < 4 cm had better five-year overall survival rates in a study by Bermudez et al. (76% vs. 18%; *p* < 0.05) [71]. 

NENs are aggressive and have a high incidence of lymph node involvement even during early-stage disease. In a study by Chen et al., positive and negative lymph nodes were associated with a progression-free survival of 6.7% and 45.9%, respectively (*p* = 0.014) [56]. They also reported that the negative and positive lymph node metastasis showed a significant difference in five-year overall survival (68% vs. 42%; *p* = 0.002) and five-year progression-free survival rates (62.6% vs. 29.3%, respectively; *p* = 0.019) [56]. In another study involving patients with small cell cervical NEC, lymph node positivity had a worse cancer-specific survival (HR: 8.832; *p* < 0.001) and overall survival (HR: 8.462; *p* < 0.001) [62]. 

Smoking is significantly associated with a worse prognosis of cervical NEC, especially for early-stage disease. In a study by Chan et al., the relationship was statistically significant in early-stage disease with a hazard ratio of 2.08 (*p* = 0.037) and was not statistically significant in combined I–IV stages with a hazard ratio of 1.82 (*p* = 0.069) [64]. The increased association between smoking and HPV infection can also explain the interrelation between smoking and cervical NEC. 

Cervical NEC represents an aggressive group of tumors, unlike treatable squamous cell carcinoma and adenocarcinoma. According to a multivariable analysis, cervical small cell NEC is associated with worse survival than adenocarcinoma (stage III: HR = 2.9 and Stage IV-HR = 4.5) and squamous cell carcinoma (Stage III: HR = 1.7 and Stage IV: HR: 3.7) of similar stages. In a study by Chen et al., the five-year survival rates of small cell carcinoma (35.7%) were worse when compared to squamous cell carcinoma (60.5%, HR: 0.55; 95% CI: 0.43–0.69%) and adenocarcinoma (69.7%, HR: 0.48; 95% CI: 0.37–0.61%) patients with early-stage and node-negative disease [60]. However, large-cell and small-cell carcinoma are managed similarly and are associated with variable survival. According to Stecklein et al., large-cell cervical NEC has better outcomes in event-free survival and overall survival than small cell NEC [65].

### 4.3. Imaging of Cervical NEN

Imaging is critical for tumor staging, while histology is essential for diagnosis but not for assessing the extent of the tumor spread [72]. Hence, imaging modalities including computed tomography (CT), magnetic resonance imaging (MRI), and ^18^F fluoro-deoxy-glucose positron emission tomography/computed tomography (^18^F-FDG PET/CT) should be considered to evaluate and stage the NEN. Currently, the imaging of NENs is based on a combination of anatomic and functional imaging. 

Although cervical NEC shows enhancement on contrast-enhanced CT, the role of CT is limited due to the small size of these lesions and the low contrast resolution. Nonetheless, combined PET and CT enable the use of low-dose CT to correlate findings on PET scans in addition to attenuation correction. In addition, the advantage of a whole-body survey in a single scan makes 18-FDG PET/CT superior to other imaging modalities. As small lesions (<5 mm) can be missed on PET scans, accompanying contrast-enhanced CT in the venous phase is preferred to improve the visualization [73]. As small cell cervical, NEC has a high predilection for lymphatic and hematogenous spread; the role of FDG PET/CT is crucial to rule out distant metastasis. The metastases from cervical NEC exhibit an avid uptake of ^18^F-FDG tracer and show SUVmax values from 2.2 to 9.6. The diagnostic accuracy, sensitivity, and specificity of PET/CT for recurrent cervical NEC are 87.9%, 94.7%, and 83.7%, respectively [35]. PET/CT also assesses the metabolic changes before the morphologic changes in response to treatment. In addition, the presence or absence of response guides the further treatment plan and predicts the prognosis. 

The surface of the neoplastic cells comprising NENs, express somatostatin receptors (SSTR), through which the somatostatin hormone acts. [74]. Octreotide and lanreotide, like somatostatin analogs, act through these receptors and help slow tumor growth and inhibit hormone secretion. Their radionuclide-labeled analogs help identify this SSTR on the tumor cells and are helpful for the diagnosis, staging, and restaging of these neoplasms. Although various radionuclides were available, such as ^111^In, ^123^I, and ^99^mTc, ^111^In-DTPA (diethylene-triaminepentaacetic acid) octreotide (octreoscan) was the only FDA approved label for SRI until recently. Somatostatin receptor imaging (SRI) was first performed in patients with carcinoids and endocrine pancreatic tumors with a gamma camera. SPECT/CT fusion technique was used to improve the anatomical localization. Despite the approval of octreoscan application, the main disadvantage is its increased physiologic uptake limiting the visualization of smaller lesions, high radiation dose, and poor resolution. 

With the enhanced application of PET scans in clinical practice, somatostatin analogs are labeled with different positron-emitting radioisotopes such as Gallium-68 and Copper-64, which the FDA has recently approved. PET/CT can provide functional (PET) and anatomic (CT) details of the tumors together [73]. PET/CT with tracer 68 Gallium DOTA-(Tyr3)-octreotide (^68^Ga-DOTATATE PET/CT) is 96% sensitive and 100% specific for SSTR positive cervical NEC [75,76]. Before SSTR-PET, Octreoscan used to be the standard imaging for tumor staging and characterization [77]. Pfeifer et al. reported that Octreoscan has a sensitivity of 87–88%, and 64-Cu DOTATATE PET/CT is 97% sensitive for detecting pathologically proven NENs. PET scans using ^68^Ga-DOTA-Somatostatin Analogs (^68^Ga-DOTANOC, ^68^Ga-DOTATOC, and ^68^Ga-DOTATATE) aid in the diagnosis and staging of NENs [78]. While ^68^Ga-DOTATATE utilizes octreotate as a peptide, ^68^Ga-DOTATOC and -NOC utilize octreotide. As a radioisotope is used in the scan, 68-Gallium DOTATATE PET/CT should be avoided during pregnancy, breastfeeding, and patients < 18 years [79]. Somatostatin receptor-based 68 Ga-tetra-azacyclododecane-tetraacetic acid (DOTA)-peptide PET/CT has also shown exciting advantages over conventional imaging in the diagnosis of NENs. Here, the somatostatin analog peptide binds to somatostatin receptors 2 and 5 with high affinity. Studies reported that 68-Gallium DOTA-peptide PET/CT is very sensitive, and 68-Ga DOTATATE PET/CT is very accurate in detecting initial or recurrent NENs [73]. 

For neoplasms > 10 mm and confined to the cervix, pelvic MRI is the imaging modality of choice to assess the size and local extension of the lesion [36]. It detects cervical NEC with a positive (PPV) and negative predictive value (NPV) of 74% and 100%, respectively, compared to PET scan, which has 44% and 44% PPV and NPV, respectively [35]. Table 4 summarizes the MRI pelvis protocol for the evaluation of gynecological tumors. The imaging features of cervical NEC on MRI are non-specific and include irregular shape and margin, adjacent structural infiltration, and lymph node involvement. Upon T2 weighted imaging, the tumor demonstrates homogeneous signal intensity alongside homogeneous enhancement compared to non-NEC cervical carcinomas. In conjunction with aggressive tumor behavior, these imaging features suggest the diagnosis of cervical NEC [48]. 

MR imaging stages cervical cancer with an accuracy ranging from 75–96%. According to a study by Xiaohui et al., the accuracy of MRI is 85.7% in staging cervical NEC [48]. It identifies vaginal and parametrial invasion with 83% and 88–97% accuracy, respectively [48]. MR imaging can ascertain parametrial involvement with 69% and 93% sensitivity and specificity, respectively [35]. Of cervical NEC cases, 37–57% involve lymph nodes, and MRI assesses the nodal involvement with a sensitivity, specificity, and accuracy of 37–90%, 71–100%, and 67–95%, respectively [35,48]. Ultrasmall supra-paramagnetic iron oxide (USPIO) particles can enhance the sensitivity of MR imaging to 93% [35]. The decreased uptake of USPIO by the lymph nodes implies metastasis. Considering 0.900 as the ADC (×10^−3^ mm^2^/s) cutoff value, diffusion-weighted imaging allows for differentiation of NEC from other cervical carcinomas with a sensitivity and specificity of 63.3% and 95%, respectively [48]. 

Positron emission tomography/magnetic resonance imaging (PET/MRI) is a hybrid imaging modality that allows for morphological evaluation of the tumor and metabolic information by the PET component [80,81]. In a recent metanalysis (n = 12), the sensitivity rate, specificity rate, diagnostic odds ratio, and area under the receiver operating characteristic curve pooled for ^18^F-FDG PET/MRI in the diagnosis of gynecological malignancies in a patient-based analysis were 74.2% (95% confidence interval, 66.2–80.8%), 89.8% (95% CI, 82.2–94.3%), 26 (95% CI, 10–67%), and 0.834, respectively, and upon lesion-based analysis were 87.5% (95% CI, 75.8–94.0%), 88.2% (95% CI, 84.2–91.3%), 50 (95% CI, 23–111%), and 0.922, respectively [82]. Another metanalysis compared the diagnostic performance of PET/CT with PET/MRI for gynecological malignancies. The study reported a slightly better diagnostic performance of ^18^F-FDG PET/MRI to that of ^18^F-FDG PET/CT at the lesion level (44 vs. 26, *p* = 0.4), as well as in the patient-level analysis (28 vs. 17, *p* = 0.48) [83].

### 4.4. Management of Cervical NEN

The treatment regimen for cervical NEC is derived from the trial designs of small cell carcinoma of the lung, as the clinical trials of the former are limited due to its low incidence. Unlike cervical SCC and adenocarcinoma, which are relatively curable even at advanced stages, NEC is highly aggressive with poor outcomes, even when detected at early stages. Hence, most treatment algorithms aim at a multimodality approach combining surgery, platinum-based chemotherapy, and radiotherapy. Table 5 and Figure 5 summarize FIGO staging, and Figure 6 shows the stage-based treatment algorithm for cervical neuroendocrine carcinoma.

#### 4.4.1. Early-Stage Tumor: IA, IB (Except IB3), and IIA1

The primary treatment, as recommended by Gynecologic Cancer InterGroup (GCIG) and the Society of Gynecologic Oncology (SGO), is radical surgery for early-stage disease and chemoradiation for an advanced-stage disease [55,84]. The optimal management for an early-stage tumor (≤4 cm) with negative lymph nodes upon imaging is radical hysterectomy and pelvic lymphadenectomy followed by chemotherapy [36]. Cohen et al. studied individuals with stages I–IIA small-cell cervical NEC and reported a significant five-year overall survival rate of 38.2% and 23.8% in individuals who did and did not undergo radical hysterectomy, respectively [85]. On the contrary, Wang et al. reported a worse five-year failure-free survival (41% vs. 60.5%; *p* = 0.086) in patients with stages IA–IIB disease who underwent surgery as primary treatment compared to those who experienced prior chemotherapy and radiotherapy [69]. As almost half of patients with small cell cervical NEC have lymph node involvement, pelvic lymphadenectomy (PLN) is included alongside radical hysterectomy. Boruta et al. and Pei et al. even recommended the routine inclusion of para-aortic lymphadenectomy in early-stage small-cell cervical NEC [54,66]. 

Adjuvant chemotherapy is recommended in early-stage disease after complete surgical resection [55]. Etoposide and cisplatin (EP) and vincristine, adriamycin, and cyclophosphamide (VAC) are the two regimens that have been advocated with improved survival compared to other regimens [54]. However, EP is preferred due to the higher toxicity of VAC. Ishikawa et al. suggested that etoposide-platinum or irinotecan-platinum are the best chemotherapy for stages I and II cervical NEC, as the response rates were 43.8% compared to 12.9% for the taxane-platinum regimen [63,86]. The number of chemotherapy cycles is also crucial for preventing the recurrence of the tumor. For instance, ≥five cycles of chemotherapy with an EP regimen following radical surgery was shown to improve five-year recurrence-free survival compared to other regimens (67.6% vs. 20.9%; *p* < 0.001) [66]. However, Yaun et al. reported improved outcomes after >4 cycles of chemotherapy, regardless of regimen, although these were statistically insignificant [51]. The SGO proposed that neoadjuvant chemotherapy (NAC) be considered when tumors are larger than 4 cm [55,56]. However, this needs validation in more and larger cohort studies. Chen et al. reported similar five-year overall survival (61% vs. 57%; *p* = 0.559) and five-year progression-free survival rates (53.5% vs. 52%; *p* = 0.511) in individuals who received and did not receive NAC [56]. Similarly, the benefit of radiotherapy in contrast to surgery is controversial for early-stage tumors. According to a study by Ishikawa et al., the hazard ratio of death was 4.74 among patients who underwent definitive radiotherapy than radical surgery [63]. In a study by Lee et al., the overall survival in patients after radical surgery alone was better than in patients after radical surgery plus adjuvant radiation (54% vs. 40%) [59]. However, the relapse rate was higher in the former group (33.3%) than in the latter (13%) in a study by Viswanathan et al. [87].

#### 4.4.2. Advanced-Stage Tumor

The SGO guidelines recommend a combination of chemotherapy and radiation in patients with locally advanced diseases and non-surgical individuals [55]. Chen et al. reported that five-year overall survival and five-year progression-free survival rates were 42.4% and 41.3% in patients who received primary radiotherapy, and 50.6% and 34.7% in patients who received primary surgery in stages IIB–IIIC2 disease, respectively [56].

Some studies have also validated the efficacy of chemotherapy among individuals with stages IIB–IV cervical NEC. Cohen et al. reported an improved three-year survival rate in individuals who did receive (17.8%) compared to those who did not receive chemotherapy (12%) either as a primary or adjuvant or with concurrent radiation [85]. An adequate number of chemotherapy cycles is essential for stage IV disease patients. Chen et al. recommended at least six cycles of chemotherapy, as it is associated with better two-year overall survival (83.3% vs. 9%, *p* < 0.001) and two-year progression-free survival (57% vs. 0%, *p* = 0.01) compared to fewer than six cycles of chemotherapy [56]. Wang et al. reported a better five-year CSS with at least five etoposide/platinum regimen cycles in 75% of stages IIB–IVB patients included in their study [69]. Immunotherapies were observed to have a therapeutic effect in high-grade extragenital NET, while their efficacy in cervical NEC has not been extensively studied. In a study including 40 specimens of cervical NEC, the authors concluded that cervical NEC is unlikely to be responsive to PD-L1 inhibitors due to the negative PD-L1 expression [88]. As the majority of cervical NECs expressed PARP-1 (poly adenosine diphosphate-ribose polymerase-1), PARP inhibitors may be included in the management plan [88]. 

## 5. Ovarian NEN

Ovarian NENs constitute 1–2% of ovarian tumors. The differences in the ovarian NEN 2014 and 2020 WHO classifications are outlined in Table 6.

### 5.1. Ovarian Carcinoid Tumors

Most ovarian NENs are benign carcinoid tumors that arise from dermoid cysts [11] (Figure 7). Ovarian carcinoids can be a primary or metastatic tumor. Primary ovarian carcinoids (POC) are more common than metastatic tumors, constituting 0.1% of the ovarian neoplasms [89]. Most POC manifest as a small, lobulated, solid, and unilateral soft tissue lesion in a dermoid cyst. Upon gross inspection, a larger POC may be visible as a yellow nodule [11]. Metastatic ovarian carcinoids can be frequently observed in patients with a history of carcinoid tumors in the midgut. They often present as bilateral, nodular masses with extensive lymphovascular and extra-ovarian involvement [11]. The five-year survival rate of ovarian carcinoid is 84% and 94% in patients with and without dermoid, respectively [90]. Morphologically POC comprises insular, trabecular, stromal, and mucinous carcinoid variants. The pathological features of these variants are described in Table 7 [11]. 

### 5.2. Ovarian NEC

Small cell ovarian carcinoma has two types—small carcinoma of the ovary, pulmonary type (SCCOPT), and small cell carcinoma of the ovary, hypercalcemic type (SCCOHT). Of the two types, SCCOPT is included under the category of miscellaneous tumors of the ovary according to the WHO 2014 classification and belongs to the NEN family, while SCCOHT is not categorized under the NEN family. It is more related to the rhabdoid-like tumor [91]. The differentiating features of SCCOPT and SCCOHT are outlined in Table 8 [91]. Small cell NECs are rare, and to date, only around 40 cases have been reported in the literature [8]. 

Large cell ovarian NECs are synonymous with undifferentiated non-small cell NECs of the ovary [92,93]. They are the rarest among ovarian NENs and aggressive tumors, with a median survival of only ten-months [93,94]. Even when diagnosed at stage-I, the average overall survival is 42 months [8,92]. Fewer than 60 cases of large-cell ovarian NECs have been reported in the literature, of which 18 patients were of the pure large-cell type [8,93]. Often, they are combined with the other germ cell and ovarian epithelial tumors. The composition of neuroendocrine components ranges from 10–90% in combined large cell NEC [95]. Histologically, they are composed of intermediate to large, round to oval pleomorphic cells arranged in solid, trabecular, or nested patterns with extensive mitoses and necrosis [95].

### 5.3. Management

Due to the rarity of ovarian NENs, the literature consists of case reports and small case series. The majority of small-cell NEC presented in peri- or post-menopausal women underwent hysterectomy with bilateral salpingo-oophorectomy and debulking as a part of the management [8]. Table 9 and Figure 8 describe FIGO staging, and Figure 9 and Figure 10 illustrate the diagnostic and treatment algorithms for ovarian NENs. In a study by Pang et al., the five-year overall survival and cancer-specific survival after surgery were 97% and 97%, respectively, in patients with early-stage ovarian NENs [94]. In cases of advanced disease, comprehensive treatment comprising surgery, chemotherapy, and radiotherapy was associated with an improved five-year overall (73%) and cancer-specific (70%) survival [94]. In a study including 11 case reports, age, AJCC stage, mode of treatment, and histological type were the significant prognostic factors for ovarian NENs [94]. 

## 6. Endometrial NEN

NENs are a rare group of tumors constituting 0.8% of endometrial malignancies [96] (Figure 11). They are rare and usually adenocarcinoma or small cell cancers with a neuroendocrine differentiation [14]. Around 56% of patients present with advanced disease with abnormal uterine bleeding or metastatic symptoms [96]. The diagnosis is based on histology and imaging, and dilatation and curettage provide adequate biopsy tissue to determine histology. Like cervical carcinoma, non-specific imaging features of endometrial NENs do not allow for histological distinction. Most endometrial cancers have similar imaging characteristics: low to moderate signal on T2WI and hypointensity relative to the hyperintense enhancing myometrium on dynamic contrast-enhanced MRI sequences. With tumor growth, the histological appearance of endometrial NEC becomes identical to FIGO grade 3 endometroid carcinoma and undifferentiated carcinoma. In indeterminate imaging and histology cases, immunohistochemistry aids in the differentiation of endometrial malignancies. Most endometrial NEC demonstrates a diffuse expression of ≥2 markers such as neuron-specific enolase, synaptophysin, chromogranin, or CD56, whereas endometroid and undifferentiated carcinomas exhibit focal positivity for neuroendocrine markers. Hence, to diagnose NEC, more than 20% of the tumor cells need to be positive for at least two of the markers [97]. The expression of p16, p53, and TTF1 by both NEC and endometroid carcinoma makes these markers unreliable and requires further studies to determine their accuracy. 

Nonetheless, when a highly aggressive endometrial tumor is diagnosed, NENs should be suspected. Table 10 and Figure 12 describe the FIGO staging, while Figure 13 demonstrates the treatment algorithm of endometrial NEN. Endometrial NECs are aggressive carcinomas associated with an overall survival of 22 and 12 months in stages I–II and III–IV, respectively [57]. According to Schlechtweg et al., the five-year and median survival for patients with endometrial NEC are 38.3% and 17 months from diagnosis. Compared to other endometrial carcinomas, NEC has an increased risk of death with a hazard ratio of 2.32 (95% CI: 1.88–2.88%) [96]. 

## 7. Vaginal NEN

Primary vaginal NEC is an extremely rare malignancy and was first reported by Scully et al. in 1984 [98]. The mean age of presentation is around 59 years, and the symptoms include vaginal discharge, leucorrhea, or metastatic symptoms [99]. Approximately 85% of patients with vaginal small cell NEC died within one year of diagnosis in a study by Gardner et al. [55]. Diagnosis requires the exclusion of metastases from other common sites of small cell NEC, such as the lungs and cervix [100]. Bing et al. reported that primary small cell vaginal NEC is histologically, immunohistochemically, and ultra-structurally similar to the pulmonary counterpart [101]. Histological findings include small, round, or oval cells with a scanty cytoplasm, fine granular unclear chromatin, absent or inconspicuous nucleoli, nuclear molding, and ill-defined cell borders. 

Table 11 and Figure 14 outline the FIGO staging, and Table 12 summarizes AJCC staging for vaginal cancer. The treatment regime lacks consensus due to the rarity of the tumor, and hence case-based therapy is recommended, similar to small cell carcinoma of the lungs (Figure 15). Chemoradiation alone or in combination with surgical excision is the routinely suggested management. Cisplatin and etoposide have been described as effective in previous studies [102]. Patients with localized upper one-third vaginal NEC may need a radical hysterectomy, partial vaginectomy, and lymphadenectomy [102]. Advanced primary small cell vaginal NEC management is through concurrent chemoradiation alongside anterior and posterior chemotherapy [103]. Currently, evidence is limited to the case reports and case series. Hence more extensive studies are required to emphasize the specific diagnosis and management plan. Figure 15 describes the treatment algorithm of vaginal/vulvar NEN.

## 8. Vulvar NEC

Primary vulvar small cell NEC resembles Merkel cell carcinoma (MCC) histologically; however, it differs in the immunohistochemistry [91]. Small cell vulvar NEC demonstrates TTF-1 uptake and MCC stains for neurofilament and CK20. Around 90% of MCC express CK20 and 30% of small vulvar cell NEC express TTF-1 uptake [104]. In addition, the detection of polyomavirus helps as a differentiating feature. With the literature being limited, most of the reported cases were found admixed with other histological types, most commonly squamous cell carcinoma and vulvar intraepithelial neoplasia [105]. FIGO staging of vulvar NEN is described in Table 13 and Figure 16. Surgical excision with or without lymphadenectomy and adjuvant chemoradiation is recommended for patients with early-stage disease, and definitive chemoradiation is recommended for advanced-stage disease [10,11] (Figure 15).

## 9. Future Perspective

Novel targeted therapies could play a vital role in treating gynecological tract NENs. Immunotherapeutic and targeted agents have been effective in other high-grade NENs [106,107,108]. In addition, the immune checkpoint inhibitors (ICIs) for programmed cell death-1 (PD-1) or programmed death-ligand 1 (PD-L1) have reported favorable response results in gynecological malignancies [109,110,111]. Other molecular targeted therapeutic agents, such as mTOR, MEK, and PTEN inhibitors, and anti-antiangiogenic inhibitors, have demonstrated encouraging results in other non-gynecological NECs [112]. Nevertheless, their part in treating gynecologic tract NENs is limited due to low incidence or insufficient clinical trials data. A comprehensive study by Mahdi et al. reported that gynecological NEC demonstrated frequent mutations in KMT2C and KMT2D genes involved in gene enhancers regulation [113]. They also found that DDR (DNA damage response) gene mutations are high in gynecological NEC, making them susceptible to PARP inhibitors. In addition, mutations in CCDC6 suggest promising benefits from PARP inhibitors [113]. Further studies are required to evaluate the efficacy of targeted immunotherapy against HMGB1/TIM-3 and CD47/SIRPα. 

## 10. Conclusions

Gynecologic tract NENs are diverse tumors and often involve a multidisciplinary management regime. The small cell NEC of the cervix and the carcinoid of the ovary are the frequent gynecologic NENs. The imaging feature of non-specific for diagnosis; nonetheless, imaging is quintessential in baseline staging, treatment response assessment, and surveillance. The adapted staging system for gynecologic tract NENs is the Revised International Federation of Gynecology and Obstetrics (FIGO) Staging System remains. There is a lack of consensus on standard treatment guidelines for gynecologic tract NENs. The treatment regimens are usually based on other gynecologic and non-gynecologic malignancies. Clinical trials, tumor genomic decoding for target therapies, accurate pathology grading, tumor bases registries, and referral to experience centers are critical elements to improve the outcome in these tumors.

## Figures and Tables

**Figure 1 cancers-14-01835-f001:**
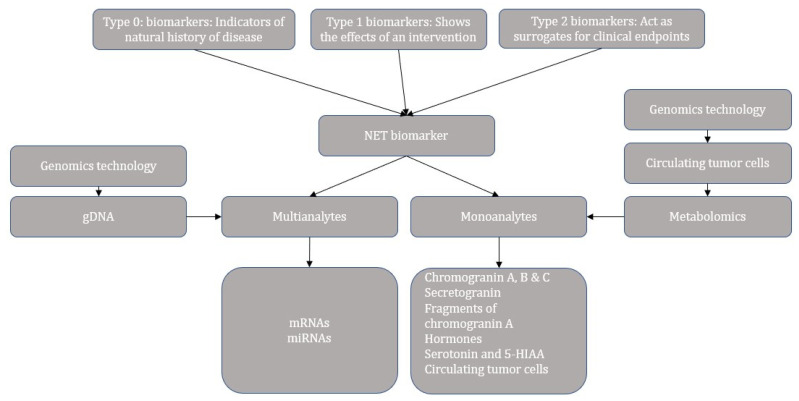
Categories and progress overview of neuroendocrine tumor biomarkers.

**Figure 2 cancers-14-01835-f002:**
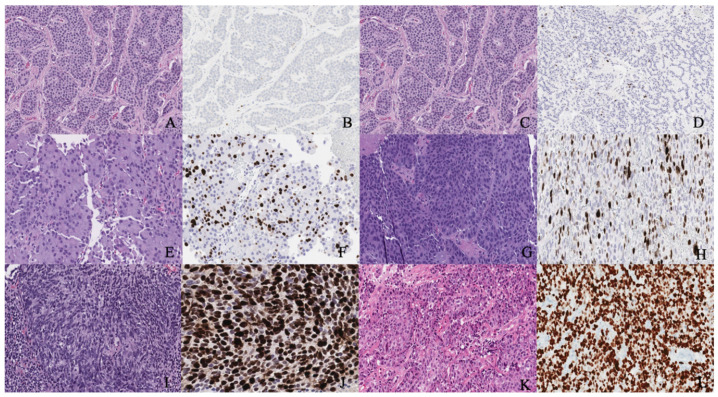
(**A**,**B**) Well-differentiated NET grade 1. (**A**) Well-differentiated neuroendocrine tumor grade 1 (low grade) with an organoid pattern, with a meshwork of thin fibrovascular septa surrounding nests of tumor cells. Tumor cells are uniform with a polygonal shape, round to oval nuclei with salt and pepper chromatin, inconspicuous nucleoli, and moderate to abundant eosinophilic cytoplasm. (**B**) The Ki67 immunohistochemical stain shows a proliferation rate of 2%. (**C**,**D**) Well-differentiated NET grade 2. (**C**) Neuroendocrine cells in a well-differentiated neuroendocrine tumor, grade 2. Tumor cells are relatively uniform and round. The nuclear chromatin is finely granular. (**D**) The Ki67 immunohistochemical stain shows a proliferation rate of 10%. (**E**,**F**) Well-differentiated NET grade 3. (**E**) Neuroendocrine cells in a well-differentiated neuroendocrine tumor, grade 3. Tumor cells are relatively uniform and round with eosinophilic cytoplasm. The nuclear chromatin is granular. (**F**) The Ki67 immunohistochemical stain shows a proliferation rate of 30–40%. (**G**,**H**) Poorly differentiated neuroendocrine carcinoma. (**G**) The tumor shows solid nests of poorly differentiated epithelioid cells with dense chromatin. (**H**) Ki67 immunohistochemical stain shows a proliferation rate of 80%. (**I**,**J**) Small cell carcinoma. (**I**) Sheets of oval blue cells with minimal cytoplasm. The chromatin is dense. Nuclei demonstrate molding and smudging. (**J**) Ki67 shows a proliferation index of 80%. (**K**,**L**) Large cell neuroendocrine carcinoma. (**K**) Tumor cells with sheets of large, epithelioid cells. Cytologic features show abundant cytoplasm, coarse chromatin, nuclear pleomorphism, and prominent nucleoli. (**L**) Ki67 is greater than 90%.

**Figure 3 cancers-14-01835-f003:**
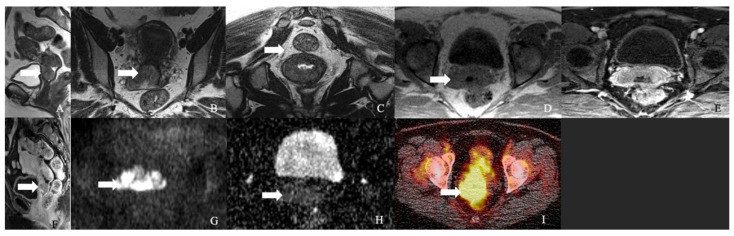
A 62-year-old female with neuroendocrine carcinoma of the cervix. (**A**) Sagittal T2 weighted image, (**B**) axial T2 weighted image, (**C**) oblique T2 weighted image, (**D**) axial pre-contrast T1 weighted image, (**E**) axial fat-saturated post-contrast T1 weighted image, (**F**) sagittal fat-saturated post-contrast T1 weighted image, (**G**) axial diffusion-weighted images (B-100), (**H**) axial apparent diffusion coefficient MRI images, and (**I**) Axial Ga-68 dotatate PET/CT image demonstrate a FDG avid enhancing mass in the cervix (arrow) with restricted diffusion and no parametrium involvement. The mass biopsy reported a small cell neuroendocrine carcinoma.

**Figure 4 cancers-14-01835-f004:**
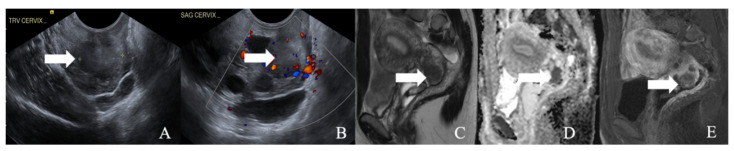
A 52-year-old female with a neuroendocrine tumor of the cervix uteri. (**A**) Transverse and (**B**) sagittal ultrasound image of the cervix demonstrates a heterogenous cervical mass (arrow) measuring about 14 cm. (**C**) Sagittal T2 weighted image, (**D**) sagittal apparent diffusion coefficient map, and (**E**) sagittal fat-saturated post-contrast T1 weighted MRI images demonstrate a mass in the cervix uteri (arrow) with restricted diffusion. The mass biopsy reported a small cell neuroendocrine tumor of grade G3.

**Figure 5 cancers-14-01835-f005:**
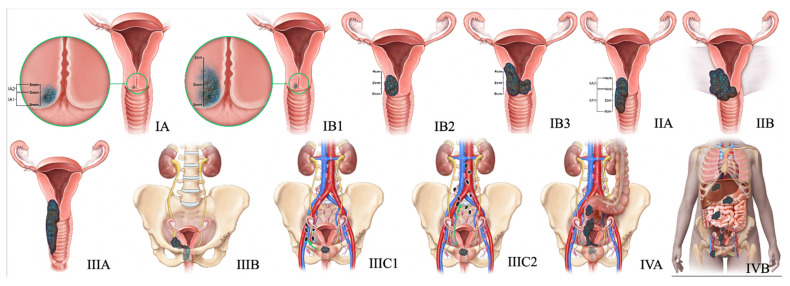
2018 FIGO staging system for cervical cancer. Stage I, confined to the cervix. Stage IA, ≤5 mm depth. Stage IA1, ≤3 mm depth. Stage IA2, 3 mm and ≤5 mm depth. Stage IB, >5 mm depth. Stage IB1, ≤2 cm maximum diameter. Stage IB2, >2 cm and ≤4 cm maximum diameter. Stage IB3, >4 cm maximum diameter. Stage II, beyond the uterus but not involving the lower one-third of the vagina or pelvic sidewall. Stage IIA, upper two-thirds of the vagina. Stage IIA1, upper two-thirds of the vagina and ≤4 cm. Stage IIA2, Upper two-thirds of the vagina and >4 cm. Stage IIB, parametrial invasion. Stage III, lower vagina, pelvic sidewall, ureters, and lymph nodes. Stage IIIA, lower one-third of the vagina. Stage IIIB, pelvic sidewall. Stage IIIC, pelvic, and para-aortic lymph node involvement. Stage IIIC1, pelvic lymph node involvement. Stage IIIC2, para-aortic lymph node involvement. Stage IV, adjacent and distant organs. Stage IVA, rectal or bladder involvement. Stage IVB, distant organs outside the pelvis.

**Figure 6 cancers-14-01835-f006:**
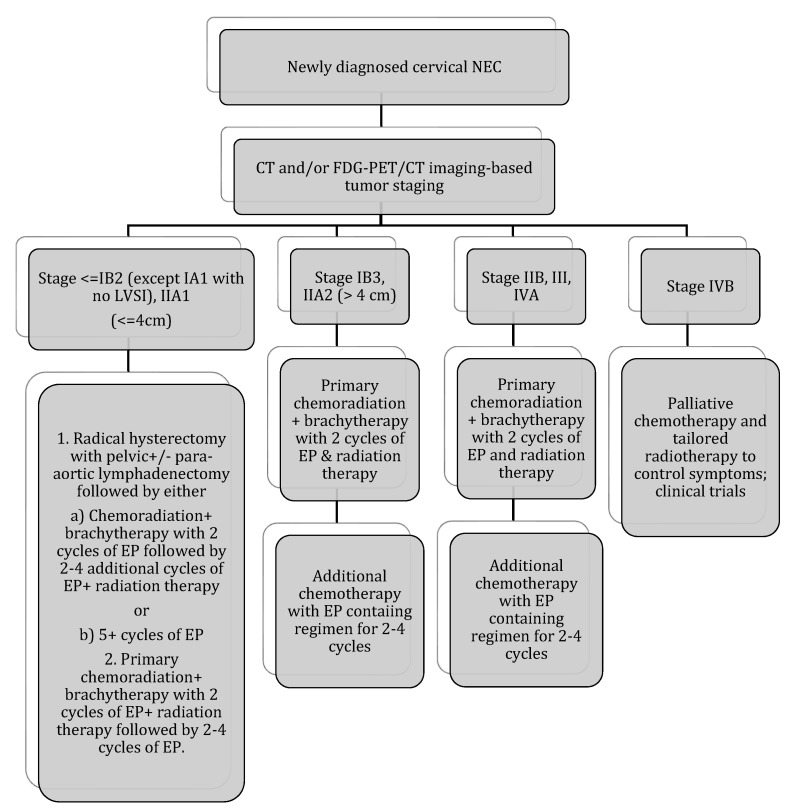
Stage-based treatment algorithm for cervical neuroendocrine carcinoma [2]. NEC: neuroendocrine carcinoma; LVSI: lymphovascular space invasion; EP: etoposide and cisplatin; chemotherapy with EP alone: cisplatin 60–80 mg/m^2^ on day 1 every 3 weeks and etoposide 80–120 mg/m^2^ on days 1–3 every 3 weeks (with growth factor support); chemotherapy with EP + radiation therapy: 2 cycles of EP q3 weeks (cisplatin 60 mg/m^2^ and etoposide 100 mg/m^2^ given on day 1 of 21 day cycle) during radiation therapy followed by an additional 2–4 cycles of EP alone (recommended regimen). Radiotherapy: 40–45 Gy external beam radiotherapy ±40–45 Gy brachytherapy (modified from Winer et al.).

**Figure 7 cancers-14-01835-f007:**
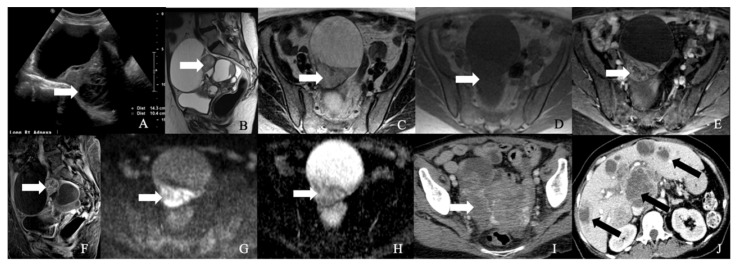
52-year-old female with a neuroendocrine tumor of the ovary. (**A**) Transverse ultrasound image of the pelvis demonstrates a heterogeneous mass in the pelvis (arrow) measuring about 14 cm. (**B**) Sagittal T2 weighted image, (**C**) axial T2 weighted image, (**D**) axial pre-contrast T1 weighted image, (**E**) axial fat-saturated post-contrast T1 weighted image, (**F**) sagittal fat-saturated post-contrast T1 weighted image, (**G**) axial diffusion-weighted images (B-100), (**H**) axial apparent diffusion coefficient images demonstrate a multiloculated mass with enhancing nodule (arrow) and restricted diffusion, (**I**) axial post-contrast CT image of the pelvis shows the heterogenous mass (arrow), and (**J**) axial post-contrast CT image of the abdomen demonstrates the multiple hepatic metastases (arrow). The mass was resected, and a biopsy reported neuroendocrine tumor of the ovary with synaptophysin, CD56, CK7, and CDX-2 were positive, and chromogranin was focally positive, with a Ki67 index of less than 5%.

**Figure 8 cancers-14-01835-f008:**
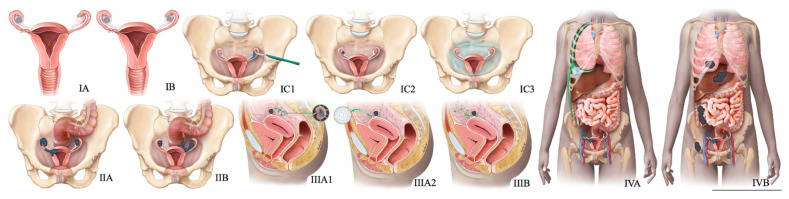
FIGO staging system for ovarian cancer. In stage IA cancer, the tumor is limited to one ovary or fallopian tube. The tumor is limited to both ovaries or fallopian tubes in stage IB cancer. Stage IC1 cancer results from the intraoperative spill. In stage IC3 cancer, malignant cells are found in ascites or peritoneal washings. In stage IIA, the tumor extends to or is implanted on (or both) uterus or fallopian tubes (or both). Stage IIIA1 cancer shows positive retroperitoneal lymph nodes. Stage IIIA2 cancer, microscopic, extrapelvic (above pelvic brim) peritoneal involvement is seen with or without positive retroperitoneal lymph nodes. Stage IIIB or stage IIIC cancer, macroscopic, extrapelvic (above pelvic brim) peritoneal involvement is seen with or without positive retroperitoneal lymph nodes (≤2 cm for stage IIIB and >2 cm for stage IIIC). Stage IVA cancer indicates pleural effusion with positive cytology. Stage IVB cancer, hepatic or splenic parenchymal metastasis is seen (or both) and metastasis to extra-abdominal organs.

**Figure 9 cancers-14-01835-f009:**
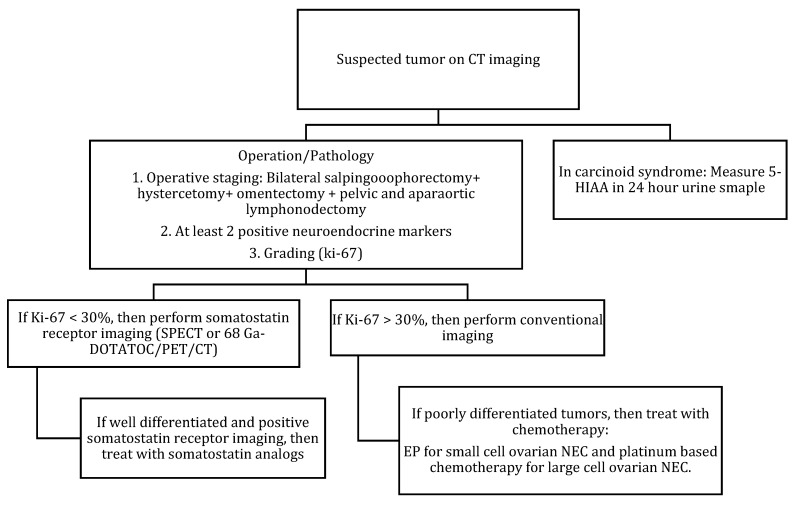
Diagnostic algorithm for ovarian NENs.

**Figure 10 cancers-14-01835-f010:**
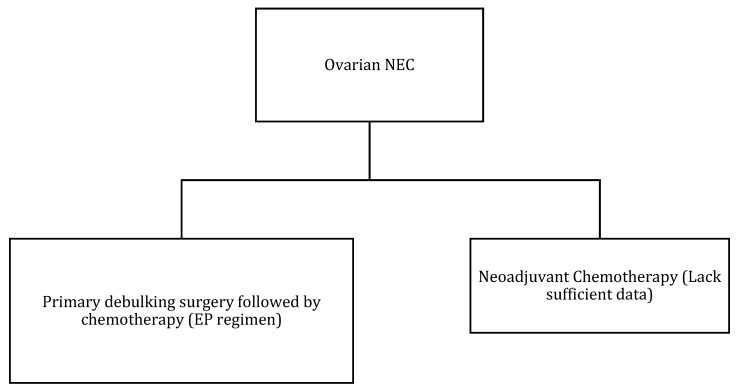
Treatment algorithm of ovarian NEC. If non-surgical or metastatic disease: palliative chemotherapy (EP regimen) or enrollment into clinical trials.

**Figure 11 cancers-14-01835-f011:**
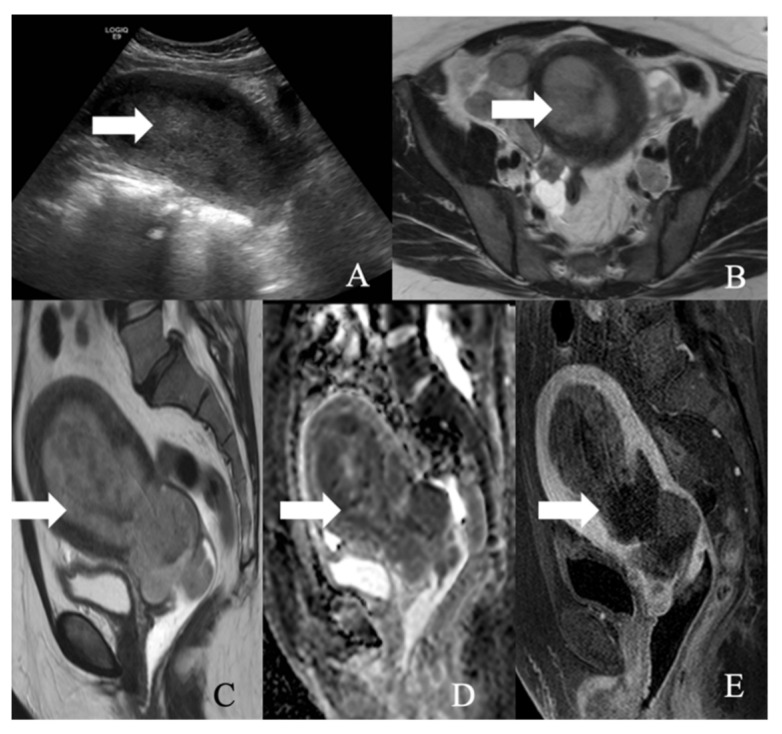
A 52-year-old female with a large cell neuroendocrine tumor of the endometrium. (**A**) Transverse ultrasound image of the uterus demonstrates thickened endometrium (arrow), (**B**) axial T2 weighted image, (**C**) sagittal T2 weighted image, (**D**) sagittal apparent diffusion coefficient map, and (**E**) sagittal fat-saturated post-contrast T1 weighted MRI images demonstrate a hypoenhancing tumor in the endometrium (arrow) with restricted diffusion.

**Figure 12 cancers-14-01835-f012:**
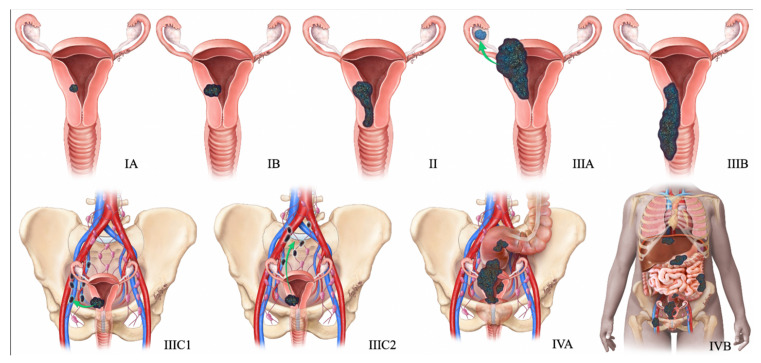
2009 FIGO staging system for endometrial cancer. Stage I tumor confined to the uterus. Stage IA < 50% myometrial invasion. Stage IB ≥ 50% myometrial invasion. Stage II Cervical stromal invasion. Stage IIIA tumor invasion into serosa or adnexa. Stage IIIB vaginal or parametrial involvement. Stage IIIC1 pelvic node involvement. Stage IIIC2 paraaortic node involvement. Stage IVA tumor invasion into bladder or bowel mucosa. Stage IVB distant metastases (including abdominal metastases) or inguinal lymph node involvement.

**Figure 13 cancers-14-01835-f013:**
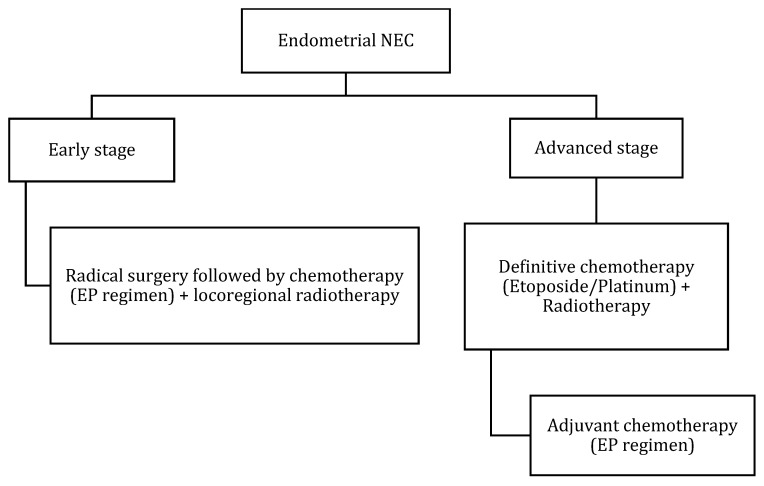
Treatment algorithm of endometrial NEC. If non-surgical or metastatic disease: palliative chemotherapy with EP regimen or enrollment into clinical trials.

**Figure 14 cancers-14-01835-f014:**
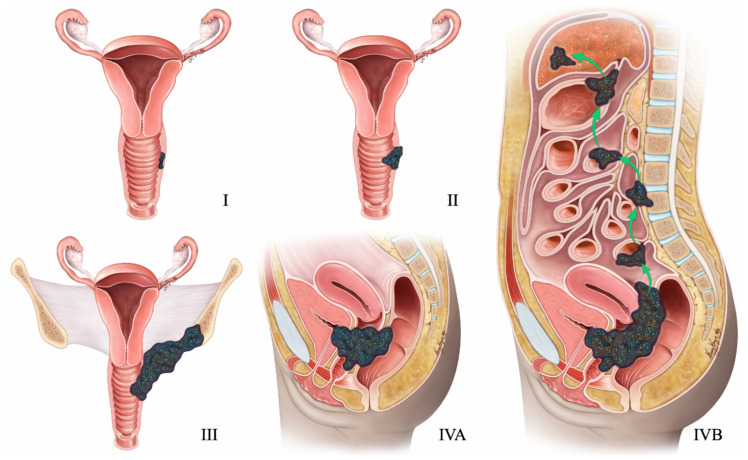
FIGO Staging of vaginal tumors. Stage I is limited to the vaginal wall. Stage II is beyond the vaginal wall without pelvic sidewall involvement. Stage III extends to the pelvic sidewall. Stage IVA infiltrates bladder or rectum or the tumor extending beyond the pelvis, with any nodal metastasis. Stage IVB has distant metastasis.

**Figure 15 cancers-14-01835-f015:**
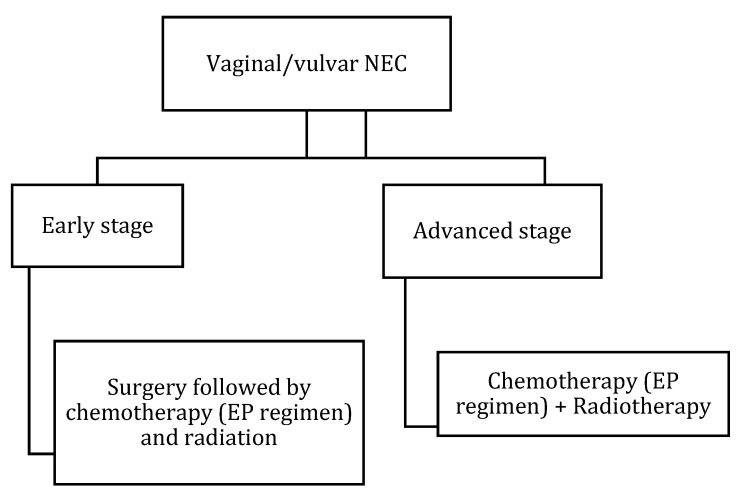
Treatment algorithm of vaginal/vulvar NEC. If non-surgical or metastatic disease: palliative chemotherapy with EP regimen or enrollment in clinical trials.

**Figure 16 cancers-14-01835-f016:**
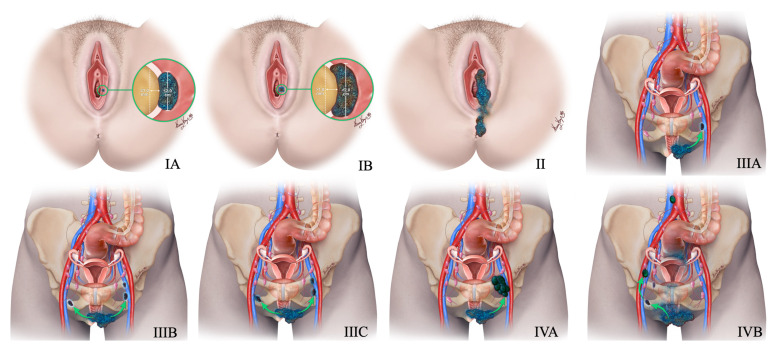
New 2021 FIGO Staging of vulvar tumors. Stage-I tumor confined to the vulva. Stage IA tumor size less than equal to 2 cm and stromal invasion less than equal to 1 mm. Stage IB tumor size more than 2 cm and stromal invasion more than 1 mm. Stage II tumor of any size with extension to lower one-third of the urethra, lower one-third of the vagina, lower one-third of the anus with negative nodes. Stage III tumor of any size with extension to the upper part of adjacent perineal structures or with any number of nonfixed, nonulcerated lymph nodes. Stage IIIA tumor of any size with disease extension to the upper two-thirds of the urethra, upper two-thirds of the vagina, bladder mucosa, rectal mucosa, or regional lymph node metastases less than equal to 5 mm. Stage IIIB regional lymph node metastases more than 5 mm. Stage IIIC regional lymph node metastases with extracapsular spread. Stage IV tumor of any size fixed to bone or fixed, ulcerated lymph node metastases, or distant metastases. Stage IVA disease fixed to the pelvic bone or fixed or ulcerated regional lymph node metastases. Stage IVB distant metastases.

**Table 1 cancers-14-01835-t001:** WHO terminology for NEN of gynecologic tract, fifth edition.

Category	Grade	Site
NET	1,2	Uterus, cervix, vulva, vagina
NEC:	3	Ovary, uterus, cervix, vulva, vagina
Small cell		
Large cell		
combined small cell NEC		
combined large cell NEC		
Carcinoid	1	Ovary only

WHO: World Health Organization; NEN: Neuroendocrine neoplasms; NET: Neuroendocrine tumors; NEC: Neuroendocrine carcinoma.

**Table 2 cancers-14-01835-t002:** In general, Grading criteria for neuroendocrine neoplasms.

Terminology	Differentiation	Grade	Mitotic Rate	Ki-67 Index
NET G1	Well-differentiated	Low	<2	<3%
NET G2	Well-differentiated	Intermediate	2–20	3–20%
NET G3	Well-differentiated	High	>20	>20%
SCNEC	Poorly differentiated	High	>20	>20%
LCNEC	Poorly differentiated	High	>20	>20%
MiNEN	Well or poorly differentiated	Variable	Variable	Variable

Mitotic index: mitotic figure count after staining with eosin–hematoxylin; Ki-67 index: percentage of Ki-67 positive cells in the tumor area with high nuclear labeling (hot spot) based on 500–2000 tumor cells.

**Table 3 cancers-14-01835-t003:** Overview of biomarkers in NEN.

Circulating and Imaging Biomarkers	Considerations	Level of Evidence
^68^Ga-SRS	It is a PET imaging with a positron-emitting radionuclide. It has been extensively studied with excellent metrics. It is considered a gold standard imaging modality in NEN. False positive rates may be observed in inflammation, renal cancer, lymphoma, and meningioma.	High
Chromogranin A	It is a component of dense core granules in neuroendocrine cells. Extensive studies are performed with varied testing subjects and results. Moderate metrics with high false-positive rates due to interaction with proton-pump inhibitors, and renal and liver failure.	Moderate
NETest	It is an multianalyte with mRNA transcripts. Prospective studies were conducted with excellent statistics. It is limited by its lower availability and higher costs.	Moderate
CT	It is an imaging modality. It has excellent availability and is less expensive. However, limited by the radiation dose and the application of nephrotoxic contrast agents.	Moderate
MRI	It is an imaging modality. It is easily available and has intermediate costs. However, limited by the duration of the scan and patient contraindications, such as claustrophobia and metal implants in the body.	Moderate
^111^In-SRS	It is a SPECT imaging with γ-emitting radionuclide. Inferior compared to excellent metric of ^68^Ga-SRS.	Moderate
^18^F-FDG	It is a PET imaging with positron-emitting radionuclide. It has low sensitivity in identifying well-differentiated NEN. Limited by its high cost.	Moderate
Chromogranin B	It is a part of family of chromogranin A. Limited studies are available with poor metrics. May be considered sensitive to rectal and ovarian NEN.	Low
Neuron-specific enolase	It is a glycolytic enzyme. Limited studies are available with poor metrics in well-differentiated NEN. It is expressed in poorly differentiated tumors.	Low
Amine uptake	It is a SPECT or PET imaging with amine transporters. It is limited by its availability (^18^F-DOPA and ^11^C-5-HTP) and sensitivity (^123^I-MIBG).	Low

SRS: Stimulated Raman Scattering; SPECT: single-photon emission computed tomography; ^18^F-DOPA: ^18^F-3,4 dihydroxyphenylalanine; ^11^C-5-HTP: ^11^C-5-hydroxy-L-tryptophan; ^123^I-MIBG: ^123^I-metaiodobenzylguanidine.

**Table 4 cancers-14-01835-t004:** MRI of the pelvis protocol.

Description	Field-of-View (FOV)	Slice Thickness	Spacing	Frequency Encoding	Frequency × Phase
Coronal T2 (include kidneys)	420	5	0	S/I	288 × 192
Sagittal T2	240	5	0	A/P	320 × 224
Field of View Sagittal b = 50, 600	240	5	0	S/I	96 × 80
Axial T2	240	5	0	L/R	320 × 224
Axial T1	240	5	0	L/R	320 × 224
Axial DWI b = 50, 400, 800	380+	5	0	L/R	96 × 160
Axial 3D Precontrast contrast	240	5	−2.5	L/R	320 × 224
Dynamic	240	5	−2.5	S/I	256 × 224
Post contrast Axial 3D immediate delay	240	5	−2.5	L/R	320 × 224

**Table 5 cancers-14-01835-t005:** FIGO 2018 staging of cervical cancer.

Stage	Extent of Disease
I	Tumor confined to cervix
IA	≤5 mm depth
IA1	≤3 mm depth
IA2	3 mm and ≤5 mm depth
IB	>5 mm depth
IB1	≤2 cm maximum diameter
IB2	>2 cm and ≤4 cm maximum diameter
IB3	>4 cm maximum diameter
Stage II	Beyond the uterus but not involving the lower one-third of the vagina or pelvic sidewall
IIA	Upper two-thirds of the vagina
IIA1	Upper two-thirds of the vagina and ≤4 cm
IIA2	Upper two-thirds of the vagina and >4 cm
IIB	Parametrial invasion
Stage III	Lower vagina, pelvic sidewall, ureters, and lymph nodes
IIIA	Lower one-third of the vagina
IIIB	Pelvic sidewall
IIIC	Pelvic and para-aortic lymph node involvement
IIIC1	Pelvic lymph node involvement
IIIC2	Para-aortic lymph node involvement
Stage IV	Adjacent and distant organs
IVA	Rectal or bladder involvement
IVB	Distant organs outside the pelvis

**Table 6 cancers-14-01835-t006:** Differences between WHO’s 2014 and 2020 classification of ovarian neuroendocrine tumors.

WHO Classification	Category	Tumor
WHO 2014	Monodermal teratoma and somatic-type tumors from a dermoid cyst	Carcinoid
	Miscellaneous tumors	Small-cell ovarian NEC pulmonary type
	Miscellaneous tumors	paraganglioma
	No category	Small-cell ovarian NEC hypercalcemia type
WHO 2020	Neuroendocrine neoplasms	Grade-1: Carcinoid; Grade-3: ovarian NEC

**Table 7 cancers-14-01835-t007:** Variants of ovarian carcinoid tumors and their pathological and immunohistochemical features.

Variant	Pathological Feature	Immunohistochemistry
Insular (most common: around 50%)	Polygonal tumor cells arranged in nests with an abundant cytoplasm accommodating eosinophilic granules, a round/oval nuclei containing salt and pepper chromatin; Hyaline appearing conspicuous stroma with sporadical psammomatous calcification.	Positive for chromogranin, synaptophysin, CD56, and CK7. Negative for CK20.
Trabecular	Parallel trabecular/wavy ribboned arrangements of regular cells with similar nuclear features to insular variant.	Positive for synaptophysin, CD56, and CK7. Negative for CK20. Negative for chromogranin similar to Sertoli cell tumor (SCT) with trabecular architecture. SCT is usually positive for inhibin and calretinin and negative for synaptophysin compared to trabecular carcinoid, which exhibits a converse immune profile.
Strumal (40% of ovarian carcinoids)	Admixture of insular/trabecular carcinoid elements and thyroid tissue; Intestinal type mucinous glands can be seen in 40% of cases.	Positive staining for carcinoid component (neuroendocrine markers) and thyroid component (thyroid transcription factor-1).
Mucinous, also known as goblet cell carcinoids (rarest)	Glands or acini lined by cuboidal/columnar epithelium with mucin-filled-cytoplasm.	CDX2 and CK20 positive; CK7 negative.

**Table 8 cancers-14-01835-t008:** Differentiating features between ovarian small cell carcinoma of pulmonary and hypercalcemic type.

Features	SCCOPT	SCCOHT
Mean age at diagnosis	51 years	24 years
Histopathological features	Spindle-shaped cells; scanty cytoplasm; inconspicuous nucleoli in sheets, dispersed chromatin, and nuclear molding	Prominent nucleoli, clumped chromatin; larger cells in 50% of cases
Laterality	Bilateral in 50% of cases	Almost always unilateral
Hypercalcemia	None	>65% of cases
Immunohistochemistry	Positive for chromogranin A in 53% of cases	Positive for chromogranin A in 9.5% of cases and vimentin in 94% of cases
Molecular features	Retained expression of SMARCA4	Loss of expression of SMARCA4

**Table 9 cancers-14-01835-t009:** The International Federation of Gynecology and Obstetrics (FIGO) staging of ovarian cancer (2014).

Stage	Extent of Disease
I	Tumor confined to ovaries
IA	Tumor limited to 1 ovary, capsule intact, no tumor on surface, negative washing
IB	Tumor involves both ovaries otherwise like IA
IC	Tumor limited to 1 or both ovaries
IC1	Surgical spill
IC2	Capsule rupture before surgery or tumor on ovarian surface
IC3	Malignant cells in the ascites or peritoneal washings
II	Tumor involves 1 or both ovaries with pelvic extension (below the pelvic brim or primary peritoneal cancer
IIA	Extension and/or implant on uterus and/or fallopian tubes
IIB	Extension to other pelvic intraperitoneal tissues
III	Tumor involves 1 or both ovaries with cytologically or histologically confirmed spread to the peritoneum outside the pelvis and / or metastatic to the retroperitoneal lymph nodes
IIIA1	Positive retroperitoneal lymph nodes only
IIIA1(i)	Metastasis ≤ 10 mm
IIIA1(ii)	Metastasis > 10 mm
IIIA2	Microscopic, extrapelvic (above the brim) peritoneal involvement ± positive retroperitoneal lymph nodes
IIIB	Macroscopic, extrapelvic, peritoneal metastasis ≤ 2 cm ± positive retroperitoneal lymph nodes. Includes extension to the capsule of liver/spleen
IIIC	Macroscopic, extrapelvic, peritoneal metastasis > 2 cm ± positive retroperitoneal lymph nodes. Includes extension to the capsule of liver/spleen
IV	Distant metastasis excluding peritoneal metastasis
IVA	Pleural effusion with positive cytology
IVB	Hepatic and/or splenic parenchymal metastasis, metastasis to extra-abdominal organs (including inguinal lymph nodes and lymph nodes outside the abdominal cavity)

**Table 10 cancers-14-01835-t010:** 2009 FIGO staging for endometrial cancer.

Stage	Extent of Disease
I	Tumor confined to uterus
IA	<50% myometrial invasion
IB	≥50% myometrial invasion
II	Cervical stromal invasion
IIIA	Tumor invasion into serosa or adnexa
IIIB	Vaginal or parametrial involvement
IIIC1	Pelvic node involvement
IIIC2	Paraaortic node involvement
IVA	Tumor invasion into bladder or bowel mucosa
IVB	Distant metastases (including abdominal metastases) or inguinal lymph node involvement

**Table 11 cancers-14-01835-t011:** 2009 FIGO staging of vaginal cancer.

Stage	Extent of Disease
I	Tumor confined to vaginal wall
II	Involvement of paravaginal tissue but not the pelvic wall
III	Either spread to pelvic lymph nodes and/or pelvic wall and/or lower vagina and/or causing hydronephrosis
IVA	Involvement of bladder, rectum or beyond pelvis with/without positive lymph node
IVB	Spread to distant organs

**Table 12 cancers-14-01835-t012:** American Joint Committee on Cancer (AJCC) staging of vaginal cancer.

AJCC Stage	Corresponding FIGO Stage	Description
IA	I	Tumor size < 2 cm and confined to the vagina
IB	I	Tumor > 2 cm and confined to the vagina
IIA	II	Tumor < 2 cm and is beyond the vaginal wall without the involvement of pelvic side wall
IIB	II	Tumor > 2 cm and is beyond the vaginal wall without pelvic sidewall involvement
III	III	Tumor extends to the pelvic side wall, lower third of the vagina, and/or causes hydronephrosis; Pelvic/inguinal lymph node metastasis
IVA	IVA	Infiltration of bladder or rectum or the tumor extending beyond the pelvis; any nodal metastasis
IVB	IVB	Distant metastasis

**Table 13 cancers-14-01835-t013:** New (2021) FIGO staging for carcinoma of the vulva.

Stage	Extent of Disease
I	Tumor confined to vulva
IA	Tumor size less than equal to 2 cm and stromal invasion less than equal to 1 mm
IB	Tumor size more than 2 cm and stromal invasion more than 1 mm
II	Tumor of any size with extension to lower one-third of the urethra, lower one-third of the vagina, lower one third of the anus with negative nodes
III	Tumor of any size with extension to upper part of adjacent perineal structures, or with any number of nonfixed, nonulcerated lymph node
IIIA	Tumor of any size with disease extension to upper two-thirds of the urethra, upper two-thirds of the vagina, bladder mucosa, rectal mucosa, or regional lymph node metastases less than equal to 5 mm^3^
IIIB	Regional lymph node metastases more than 5 mm
IIIC	Regional lymph node metastases with extracapsular spread
IV	Tumor of any size fixed to bone, or fixed, ulcerated lymph node metastases, or distant metastases
IVA	Disease fixed to pelvic bone, or fixed or ulcerated regional lymph node metastases
IVB	Distant metastases

Depth of invasion is measured from the basement membrane of the deepest, adjacent, dysplastic, tumor-free rete ridge (or nearest dysplastic rete peg) to the deepest point of invasion. Regional refers to inguinal and femoral lymph nodes.

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
