# Peer review of "Neuroendocrine Neoplasms of the Gynecologic Tract"

_cancers, 2022, doi:10.3390/cancers14071835_

Round 1

Reviewer 1 Report

Overall very well written. This is a rare, understudied section of gynecologic oncology.  This review is comprehensive and helpful. It covers all information from molecular factors to treatment options.  The authors reviewed the current information on NEN but note that further data is needed which is true.

I don't think that the staging for each type of cancer needs to be included since that is well known/ easy to find online and takes up a lot of space.

Author Response

Response to Reviewer 1 Comments

Reviewer 2 Report

The paper by Virarkar et al. is a large review of Neuroendocrine Neoplasms of the Gynecologic Tract. This review, mainly focused on cervical and ovarian neuroendocrine neoplasm (NEN), provides useful actualization on nomenclature, biological markers and imaging for accurate clinical staging. The work is well documented, well written, and represents a useful update on the topic.

As minor remarks

  • In the paragraph “Histology & Immunohistochemistry of NEN (p3), it is stated that “immunohistochemistry can distinguish between primary and metastatic NENs. For instance, TTF1 staining indicates the primary NEN in the thyroid and lung… And later (line 198) it is stated that 71% of cervical NEN express TTF1. The same remark could be made for other markers. Therefore, a less categorical statement such as “immunohistochemistry can help to distinguish…” would be more adequate.
  • In the paragraph “Imaging of cervical NEN, the first sentence indicates that “biopsy findings may be inconclusive due to tumor heterogeneity or inadequate tissue sampling”. This is true but imaging procedures, introduced in the second sentence, cannot replace histology for the primary diagnosis of cervical cancer. Therefore, as stated in the conclusion (lines 665-666), the introduction of this paragraph should stress on the crucial role of imaging for tumor staging whereas histology is important for diagnosis but not for the evaluation of tumor spread.
  • Imaging modalities that can be used for tumor staging are very well documented in this review. However, for a non-specialized reader, it is difficult to get to what extent the expression of neuroendocrine tumor markers may enhance the sensitivity and/or specificity of imaging procedures. Is-it the case? If yes, this could be more clearly indicated. The figure 5 is interesting and would be improved if the type of imaging approach for tumor staging could be mentioned in the box “Imaging-based tumor staging” according to clinical presentation.
  • Recent data using NGS approach provide detailed molecular characterization of cervical and gynecological small cell carcinoma. Some of these result point to biological pathways that can be targeted by innovative therapies. These works deserve to be mentioned, at least in the very short “future perspective” paragraph. As possible references, Xing et al. Am J Surg Pathol 2018, Schultheis et al. Molecular Pathology 2022, Mahdi et al., Molecular Pathology 2021.
  • The paragraph on endometrial neuroendocrine carcinoma is rather elusive whereas molecular analyses have shown the heterogeneity of endometrial carcinomas. The differential diagnosis of neuroendocrine carcinoma VS high grade endometrial carcinoma or undifferentiated carcinoma may be difficult. This could be underlined and the main features that characterize these respective entities recalled (see for instance Bartosh et al., Adv Anat Pathol 2011).
  • There are some editing corrections to perform. For instance, line 209, “associated with poor (worse?) prognosis than; line 213 “as the (most?) significant …; line 241 “had a (more?) prolonged …than; line 262 “with poor (worse?) survival than…” etc

Author Response

Response to Reviewer 2 Comments
